# The risk of ischemic stroke significantly increases in individuals with blepharitis: A population-based study involving 424,161 patients

**Jing-Xing Li**[1,2,3], **Shu-Bai Hsu**[4,5], **Ying-Hsiu Shih**[6], **Yi-Yu Tsai**[2,7,8], **Ying-Hsuen Wu**[7], **You-Ling Li**[2,7], **Chun-Chi Chiang**[2,7,8]*

1 Department of General Medicine, China Medical University Hospital, Taichung, Taiwan, 2 School of Medicine, China Medical University, Taichung, Taiwan, 3 Graduate Institute of Clinical Laboratory Sciences and Medical Biotechnology, National Taiwan University, Taipei, Taiwan, 4 College of Medicine, China Medical University, Taichung, Taiwan, 5 Department of Nursing, China Medical University Hospital, Taichung, Taiwan, 6 Institute of Public Health, China Medical University Hospital, Taichung, Taiwan, 7 Department of Ophthalmology, China Medical University Hospital, Taichung, Taiwan, 8 Department of Optometry, Asia University, Taichung, Taiwan

* elsa10019@yahoo.com.tw

**Data Availability Statement:** Data of this study are available from the National Health Insurance Research Database (NHIRD) published by Taiwan National Health Insurance (NHI) Administration.

## Abstract

### Introduction

To investigate the association of blepharitis and ischemic stroke.

### Methods

This nationwide retrospective cohort study used population-based data in Taiwan. Individuals aged 20 and above with diagnosis of blepharitis was included based on electrical medical records. After exclusion of ineligible cases, 424,161 patients were identified between 2008 and 2018. The blepharitis and non-blepharitis cohorts were matched based on sex, age, and comorbidities. Multivariable-adjusted Cox proportional hazards model was adopted to calculate the hazard ratio and 95% confidence interval (CI) between blepharitis and non-blepharitis cohorts. The incidence of ischemic stroke was estimated by Kaplan–Meier analysis.

### Results

424,161 pairs of blepharitis cohort and non-blepharitis cohort were 1:1 propensity score matched for statistical analysis. Patients with blepharitis had significantly increased risk of ischemic stroke compared with the individuals without blepharitis (adjusted hazard ratio 1.32, 95% CI 1.29–1.34, $P < 0.001$). A significantly higher risk of ischemic stroke was observed in blepharitis cohort with a previous diagnosis of cancer than in those without cancer ($P$ for interaction < 0.0001). Kaplan–Meier survival analysis revealed the cumulative incidence of ischemic stroke increased in the blepharitis cohort compared with that in the non-blepharitis cohort in 10 years (log-rank $P < 0.001$). The follow-up period analysis further

Due to the implementation of the "Personal Information Protection Act" by the Taiwanese government in 2012, the data used in this study cannot be disclosed in the paper, supplemental files, or any public repository." Requests for data can be sent as a formal proposal to the NHIRD Office (https://nhird.nhri.edu.tw/en/index.html) or by email to stsung@mohw.gov.tw. Requests to access these datasets should be directed to stsung@mohw.gov.tw.

**Funding:** The author(s) received no specific funding for this work.

**Competing interests:** The authors have declared that no competing interests exist.

**Abbreviations:** aHR, adjusted hazard ratio; DM, diabetic mellitus; NSAIDs, non-steroidal anti-inflammatory drugs; HR, hazard ratio; ILs, interleukins; IS, ischemic stroke; TNF-α, tumor necrosis factor alpha; TGF-β, transforming growth factor beta; 95% confidence interval.

indicated 1.41-fold adjusted hazard (95% CI 1.35−1.46, $P < 0.001$) of ischemic stroke within a year after blepharitis diagnosis.

## Conclusions

Patients with blepharitis had an elevated risk of developing ischemic stroke. Early treatment and active surveillance are suggested for patients with chronic blepharitis. Further research is required to determine the casual relationship between blepharitis and ischemic stroke, as well as the underlying mechanism.

## Introduction

The prevalence of blepharitis varies from 37% to 50% according to several epidemiologic studies, most often affecting female population aged >50 years [1, 2]. Blepharitis, which is defined by inflammation of the eyelids, is a prevalent ocular condition that is on the rise. The inflammatory condition is characterized by foreign-body sensation, burning or stinging, light sensitivity, and eyelid crusting. It can be acute or chronic, with the latter being more prevalent. Blepharitis can be classified based to its anatomical location as anterior or posterior, however many cases overlap [3]. Anterior blepharitis is induced by staphylococcal infection or is associated with seborrheic dermatitis on the face. It appears to be more of a chronic infection. Meanwhile, posterior blepharitis is related to meibomian gland dysfunction because meibomian gland secretion function is impaired. The precise pathogenesis is thought to be multifactorial, including inflammatory skin conditions and chronic infection with bacterial, viral, or parasitic (Demodex mites) infections on the eyelids.

Stroke is a heterogeneous disease with different etiological subtypes [4]. Pathological subtypes comprise ischemic stroke (IS) and hemorrhagic stroke. IS accounts for over 80% of cerebrovascular accident occurrences. Age, sex, and race/ethnicity are nonmodifiable risk factors for IS, while hypertension, hyperlipidemia, smoking, diet, and physical inactivity are among some of the more commonly reported modifiable risk factors [5]. Recently, more and more risk factors and triggers of stroke was demonstrated, including inflammatory disorders [6, 7], infection [8], cardiovascular diseases [9−11], and air pollution [12]. Nonetheless, the link between stroke and ocular diseases have not yet been investigated.

This study aims to evaluate matched cohorts of patients with and without blepharitis to estimate the risk of IS in patients with blepharitis.

## Methods

### Study population

More than 99% population in Taiwan joint National Health Insurance (NHI) plan as a mandatory insurance system. The NHI Research Database (NHIRD) contains information about the insured, including age, sex, place of residence, income, diagnosis, medications, and medical procedures. Numerous renowned studies employed this database in past two decades [13−15]. The NHIRD encrypts each patient's identifying information in the database to ensure patient privacy. Inpatient and outpatient diagnoses are established using the International Classification of Diseases, Ninth and Tenth Revisions, Clinical Modification (ICD-9-CM and ICD-10-CM, respectively). This study was conducted following the principles of the Declaration of Helsinki. Furthermore, the study was approved by both the National Health Insurance

Administration and institutional review board of China Medical University Hospital. Since identifiable information of the patients was encrypted before release, the need for informed consent was waived by these institutions.

## Study design

This study compared the blepharitis cohort and the non-blepharitis cohort to analyze the results. Patients with blepharitis were enrolled in the blepharitis cohort between 2008 and 2018. In contrast, participants in the non-blepharitis cohort did not have a documented diagnosis of blepharitis. Diagnoses of blepharitis were based on ICD codes (ICD-9-CM 373.0; ICD-10-CM H01.0x) for at least two outpatient visits or one hospitalization record. The inclusion criteria for this study were as follows: (1) aged ≥20 years and (2) diagnosed with blepharitis. The exclusion criteria were as follows: (1) aged <20 years; (2) previous index date before January 1, 2008, or after December 31, 2018; (3) history of stroke, transient ischemia attack, graft-versus-host disease, human immunodeficiency virus infection, Sjögren's syndrome, rheumatoid arthritis, or current keratoconjunctivitis before January 1, 2008; (4) use of systemic medications related to dry eye 4 weeks before the index date, including antiandrogen, 13-cis-retinoid acid, antihistamines, lipid-lowering drugs, or β-blockers and topical medications related to dry eye 4 weeks before the index date such as ophthalmic glaucoma medications, ophthalmic steroids, ophthalmic epinephrine, ophthalmic anti-infectives, ophthalmic non-steroidal anti-inflammatory drugs (NSAIDs), artificial tears, and lubricants; systemic antiallergics (antihistamines), and anticholinergics; (5) any ocular surgery (including cataract surgery) 6 months before the index date; (6) patients diagnosed with stroke before the index date; and (7) basic information of the cases was missing. The study is conducted in accordance with the Strengthening the Reporting of Observational Studies in Epidemiology (STROBE) guidelines for reporting observational studies [16].

## Procedures

The index date for the case was the date of the initial diagnosis of blepharitis, while the index date for the control was a date chosen at random between 2008 and 2018. All study participants were followed from the index date until the onset of ischemic stroke, withdrawal from the NHI program, or the end of 2019.The variates were assessed and matched between the blepharitis cohort and the non-blepharitis cohort, including age, sex, comorbidities of hypertension, diabetic mellitus (DM), dyslipidemia, Parkinson's disease, coronary artery disease, congestive heart failure, acute myocardial infarction, non-valvular atrial fibrillation, chronic kidney disease, chronic liver disease, peripheral vascular disease, venous thromboembolism, malignancy, current cigarette smoking, and obesity.

## Main outcomes

During the follow-up, we observed and compared the incidence rates and hazard ratio (HR) of IS between the study and control groups. The development of IS was the main outcome in the present study. Any IS that occurred prior to blepharitis was not considered an outcome, we only evaluate new onset of IS. Diagnoses of IS was based on ICD codes (ICD-9-CM 433, 434, 435, 436, 437; ICD-10-CM I63, I67.89) for at least two outpatient visits or one hospitalization record.

## Statistical analysis

We used 1:1 propensity score matching of sex, age, and comorbidities to optimize the multiple variates between the blepharitis and non-blepharitis cohorts. The closest propensity score

**Table 1. Comparison of non-blepharitis and blepharitis cohorts.**

| Variables | Non-blepharitis (N = 424,161) | | Blepharitis (N = 424,161) | | |
| | n | % | n | % | SMD |
|---|---|---|---|---|---|
| Sex | | | | | |
| Female | 211114 | 49.77 | 261365 | 61.62 | 0.24 |
| Male | 213047 | 50.23 | 162796 | 38.38 | 0.24 |
| Age | | | | | |
| 20–39 | 119222 | 28.11 | 119097 | 28.08 | 0.001 |
| 40–59 | 151140 | 35.63 | 150828 | 35.56 | 0.002 |
| ≥60 | 153799 | 36.26 | 154236 | 36.36 | 0.002 |
| Mean (SD) | 52.03 | 17.74 | 52.09 | 17.77 | 0.004 |
| Comorbidities | | | | | |
| Hypertension | 120170 | 28.33 | 119845 | 28.25 | 0.002 |
| Diabetes mellitus | 58278 | 13.74 | 58041 | 13.68 | 0.002 |
| Dyslipidemia | 94292 | 22.23 | 93901 | 22.14 | 0.002 |
| Parkinson's disease | 1402 | 0.33 | 1459 | 0.34 | 0.002 |
| Coronary artery disease | 46517 | 10.97 | 46423 | 10.94 | 0.001 |
| Congestive heart disease | 9279 | 2.19 | 9310 | 2.19 | 0.000 |
| Acute myocardial infarction | 1403 | 0.33 | 1354 | 0.32 | 0.002 |
| Non-valvular atrial fibrillation | 4090 | 0.96 | 4126 | 0.97 | 0.001 |
| Chronic kidney disease | 8407 | 1.98 | 8368 | 1.97 | 0.001 |
| Chronic liver disease | 3601 | 0.85 | 3606 | 0.85 | 0.000 |
| Peripheral vascular disease | 5483 | 1.29 | 5510 | 1.30 | 0.001 |
| Malignancy | 16608 | 3.92 | 16656 | 3.93 | 0.001 |
| Venous thromboembolism | 2006 | 0.47 | 1921 | 0.45 | 0.003 |
| Smoking | 6909 | 1.63 | 4363 | 1.03 | 0.052 |
| Obesity | 3013 | 0.71 | 2886 | 0.68 | 0.004 |
| Follow-up period, mean (SD) | | | | | |
| Ischemic stroke | 7.01 | 3.34 | 7.02 | 3.32 | 0.004 |

SD, standard deviation; SMD, standardized mean difference.

between the case and control groups was estimated. Covariates such as sex, age, and comorbidities were included as independent variables (Table 1). We constructed matched pairings using the nearest-neighbor algorithm, assuming that a $P < 0.05$ indicates a significant difference between the case and control groups. Using crude and multivariable-adjusted Cox proportional hazards models, the outcomes of the blepharitis and non-blepharitis cohorts were compared. The results are reported as HR and 95% confidence interval (CI). Patients were censored on the date of respective results, death, or December 31, 2018, whichever occurred first. During the follow-up period, the Kaplan–Meier survival analysis and log-rank tests were utilized to compare the cumulative incidence of IS between the blepharitis and non-blepharitis cohorts. Statistical analysis was performed on SAS (version 9.5; SAS Institute, Cary, NC, USA) software, and a two-tailed $P < 0.05$ was denoted as significant.

## Results

From January 1, 2008, to December 31, 2019, we identified 31,488,321 patients in the database. 733,436 of these individuals had blepharitis, while 30,754,885 did not (Fig 1). After removing ineligible cases, 424,161 pairs of patients with blepharitis and non-blepharitis individuals were

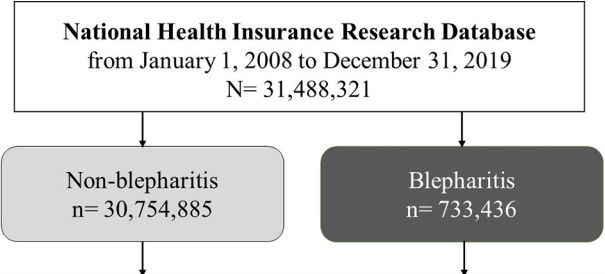

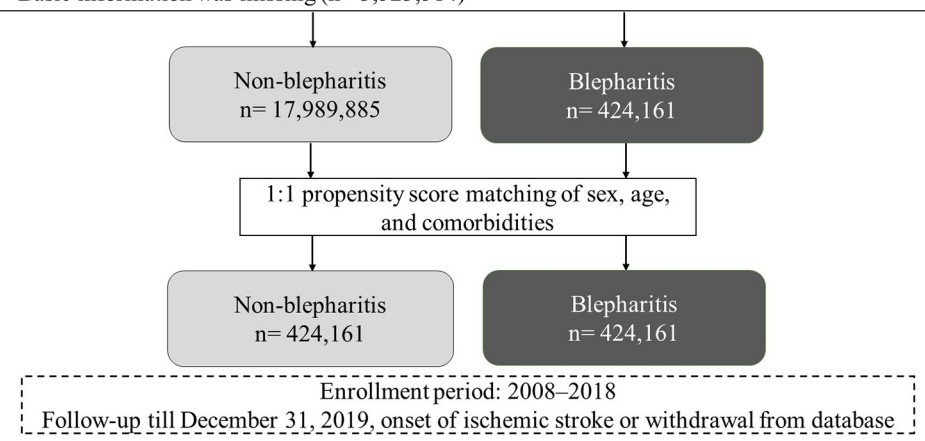

**Fig 1. Study flow chart.**

constructed using 1:1 propensity score matching. Table 1 presents the baseline characteristics of blepharitis and non-blepharitis cohorts. In blepharitis cohort, 61.62% of blepharitis patients were female, and the mean age of blepharitis cohort was 52.09 ± 17.77 years. The average period of follow-up for blepharitis patients developing IS was 7.02 ± 3.32 years.

Table 2 shows the risk factors of ischemic stroke using multivariable-adjusted Cox model. Multivariable analysis revealed 27,108 (6.40%) patients with blepharitis and 20,804 (4.90%) patients without blepharitis developing IS during the follow-up period (incidence rate, 9.10 and 7.00 per 10,000 person-years, respectively). The blepharitis cohort showed significantly increased risk of IS compared with the non-blepharitis cohort (adjusted HR [aHR] 1.32, 95% CI 1.29–1.34, $P < 0.001$). Men had significantly higher risk of IS than women (aHR 1.17, 95% CI 1.15–1.19, $P < 0.001$). The risk of IS also dramatically rose by age, for individuals aged 60 years or above, the risk of IS was 22.04-fold than age 20–39 (95% CI 20.73–23.43, $P < 0.001$).

Table 3 shows the results of stratified analysis. The effect of blepharitis on IS was significantly different between sex and age groups. In the blepharitis group with a previous diagnosis of cancer, the effect of blepharitis on IS was considerably greater than in those without cancer

**Table 2. Risk factors of ischemic stroke.**

| Variables | Ischemic stroke | | | | | | |
|---|---|---|---|---|---|---|---|
| | n | PY | IR | cHR | (95% CI) | aHR[†] | (95% CI) |
| Non-blepharitis | 20804 | 2973784 | 7.00 | 1.00 | (Reference) | 1.00 | (Reference) |
| Blepharitis | 27108 | 2979191 | 9.10 | 1.3 | (1.28, 1.32)*** | 1.32 | (1.29, 1.34)*** |
| Sex | | | | | | | |
| Female | 25174 | 3355282 | 7.50 | 1.00 | (Reference) | 1.00 | (Reference) |
| Male | 22738 | 2597693 | 8.75 | 1.16 | (1.14, 1.18)*** | 1.17 | (1.15, 1.19)*** |
| Age | | | | | | | |
| 20–39 | 1109 | 1809152 | 0.61 | 1.00 | (Reference) | 1.00 | (Reference) |
| 40–59 | 9636 | 2231548 | 4.32 | 7.01 | (6.59, 7.46)*** | 6.32 | (5.94, 6.73)*** |
| ≥60 | 37167 | 1912275 | 19.44 | 30.6 | (28.85, 32.51)*** | 22.04 | (20.73, 23.43)*** |
| Comorbidities | | | | | | | |
| Hypertension | 26983 | 1432884 | 18.83 | 3.87 | (3.8, 3.94)*** | 1.59 | (1.56, 1.63)*** |
| Diabetes mellitus | 12533 | 667764 | 18.77 | 2.65 | (2.6, 2.7)*** | 1.26 | (1.23, 1.29)*** |
| Dyslipidemia | 14957 | 1100615 | 13.59 | 1.87 | (1.84, 1.91)*** | 0.83 | (0.81, 0.84)*** |
| Parkinson's disease | 363 | 12840 | 28.27 | 3.16 | (2.85, 3.51)*** | 1.45 | (1.3, 1.6)*** |
| Coronary artery disease | 10928 | 518421 | 21.08 | 2.91 | (2.85, 2.97)*** | 1.2 | (1.17, 1.23)*** |
| Congestive heart disease | 2520 | 86626 | 29.09 | 3.4 | (3.27, 3.54)*** | 1.26 | (1.21, 1.32)*** |
| Acute myocardial infarction | 242 | 12132 | 19.95 | 2.2 | (1.94, 2.5)*** | 0.8 | (0.7, 0.91)*** |
| Non-valvular atrial fibrillation | 1331 | 37680 | 35.32 | 4.05 | (3.84, 4.28)*** | 1.5 | (1.42, 1.59)*** |
| Chronic kidney disease | 1774 | 75459 | 23.51 | 2.67 | (2.55, 2.8)*** | 1.13 | (1.08, 1.19)*** |
| Chronic liver disease | 486 | 34377 | 14.14 | 1.6 | (1.46, 1.75)*** | 0.86 | (0.79, 0.94)** |
| Peripheral vascular disease | 1062 | 56060 | 18.94 | 2.18 | (2.05, 2.32)*** | 1.04 | (0.97, 1.1) |
| Malignancy | 2609 | 186679 | 13.98 | 1.7 | (1.63, 1.76)*** | 0.94 | (0.91, 0.98)** |
| Venous thromboembolism | 286 | 19441 | 14.71 | 1.66 | (1.48, 1.87)*** | 0.85 | (0.76, 0.96)** |
| Smoking | 315 | 50205 | 6.27 | 0.69 | (0.62, 0.77)*** | 0.84 | (0.75, 0.94)** |
| Obesity | 235 | 32285 | 7.28 | 0.83 | (0.73, 0.95)** | 0.87 | (0.76, 0.99)* |

PY, person-years; IR, incidence rate, per 10,000 person-years; cHR, crude hazard ratio; aHR, adjusted hazard ratio.

[†] Adjusted hazard ratio in multivariable analysis of sex, age, and comorbidities.

* $p < 0.05$

**$p < 0.01$

***$p < 0.001$.

(aHR 1.51 versus 1.31, $P$ for interaction $< 0.0001$). Blepharitis cohort with sarcoidosis also had higher risk of IS than without sarcoidosis, whereas not significant (aHR 3.59, 95% Cl 0.26–49.92, $P = 0.3409$).

Fig 2 depicts the Kaplan–Meier curves of the cumulative incidence of IS in the blepharitis cohort and non-blepharitis cohort. The cumulative incidence of IS increased in the group with blepharitis compared to the individuals without blepharitis in 10-year follow-up (log-rank $P < 0.001$).

Table 4 presents the risks of IS in the blepharitis cohort relative to the non-blepharitis cohort in terms of follow-up time. With respect to follow-up period <1 year, the aHR of IS development in blepharitis cohort was 1.41 (95% CI 1.35–1.46, $P < 0.001$). The risk of IS slightly decreased by time but were still significantly higher in the blepharitis cohort over the non-blepharitis cohort after more than 6 years of follow-up (aHR 1.17, 95% CI 1.12–1.22, $P < 0.001$).

**Table 3. Risk stratification for ischemic stroke in non-blepharitis and blepharitis cohorts.**

| Variables | Non-blepharitis | | | Blepharitis | | | Crude | | | Adjusted | | | P for interaction |
|---|---|---|---|---|---|---|---|---|---|---|---|---|---|
| | n | PY | IR | n | PY | IR | cHR | CI | P | aHR[†] | CI | P | |
| Sex | | | | | | | | | | | | | <0.0001 |
| Female | 10038 | 1489089 | 6.74 | 15136 | 1866193 | 8.11 | 1.21 | (1.18, 1.24)*** | <0.001 | 1.29 | (1.25, 1.32)*** | <0.001 | |
| Male | 10766 | 1484695 | 7.25 | 11972 | 1112998 | 10.76 | 1.47 | (1.44, 1.51)*** | <0.001 | 1.35 | (1.31, 1.39)*** | <0.001 | |
| Age | | | | | | | | | | | | | 0.0036 |
| 20–39 | 475 | 897781 | 0.53 | 634 | 911371 | 0.70 | 1.32 | (1.17, 1.48)*** | <0.001 | 1.41 | (1.25, 1.59)*** | <0.001 | |
| 40–59 | 4316 | 1114056 | 3.87 | 5320 | 1117492 | 4.76 | 1.23 | (1.18, 1.28)*** | <0.001 | 1.26 | (1.21, 1.31)*** | <0.001 | |
| ≥60 | 16013 | 961947 | 16.65 | 21154 | 950328 | 22.26 | 1.33 | (1.3, 1.36)*** | <0.001 | 1.33 | (1.3, 1.36)*** | <0.001 | |
| Comorbidities | | | | | | | | | | | | | |
| Hypertension | | | | | | | | | | | | | <0.0001 |
| No | 8185 | 2263600 | 3.62 | 12744 | 2256492 | 5.65 | 1.56 | (1.52, 1.6)*** | <0.001 | 1.62 | (1.58, 1.67)*** | <0.001 | |
| Yes | 12619 | 710184 | 17.77 | 14364 | 722699 | 19.88 | 1.12 | (1.1, 1.15)*** | <0.001 | 1.12 | (1.09, 1.15)*** | <0.001 | |
| Diabetes mellitus | | | | | | | | | | | | | <0.0001 |
| No | 14957 | 2644011 | 5.66 | 20422 | 2641200 | 7.73 | 1.37 | (1.34, 1.39)*** | <0.001 | 1.39 | (1.36, 1.42)*** | <0.001 | |
| Yes | 5847 | 329773 | 17.73 | 6686 | 337991 | 19.78 | 1.12 | (1.08, 1.16)*** | <0.001 | 1.12 | (1.08, 1.16)*** | <0.001 | |
| Dyslipidemia | | | | | | | | | | | | | <0.0001 |
| No | 13801 | 2424951 | 5.69 | 19154 | 2427408 | 7.89 | 1.38 | (1.35, 1.42)*** | <0.001 | 1.4 | (1.37, 1.44)*** | <0.001 | |
| Yes | 7003 | 548833 | 12.76 | 7954 | 551782 | 14.42 | 1.13 | (1.1, 1.17)*** | <0.001 | 1.14 | (1.1, 1.17)*** | <0.001 | |
| Parkinson's disease | | | | | | | | | | | | | 0.3732 |
| No | 20640 | 2967426 | 6.96 | 26909 | 2972708 | 9.05 | 1.3 | (1.28, 1.32)*** | <0.001 | 1.32 | (1.29, 1.34)*** | <0.001 | |
| Yes | 164 | 6358 | 25.80 | 199 | 6483 | 30.70 | 1.18 | (0.96, 1.45) | 0.1259 | 1.12 | (0.91, 1.38) | 0.2948 | |
| Congestive heart disease | | | | | | | | | | | | | <0.0001 |
| No | 15714 | 2717460 | 5.78 | 21270 | 2717094 | 7.83 | 1.35 | (1.32, 1.38)*** | <0.001 | 1.38 | (1.35, 1.41)*** | <0.001 | |
| Yes | 5090 | 256324 | 19.86 | 5838 | 262097 | 22.27 | 1.13 | (1.08, 1.17)*** | <0.001 | 1.12 | (1.08, 1.16)*** | <0.001 | |
| Acute myocardial infarction | | | | | | | | | | | | | 0.0830 |
| No | 20687 | 2967826 | 6.97 | 26983 | 2973017 | 9.08 | 1.3 | (1.28, 1.32)*** | <0.001 | 1.32 | (1.29, 1.34)*** | <0.001 | |
| Yes | 117 | 5958 | 19.64 | 125 | 6174 | 20.25 | 1.03 | (0.8, 1.33) | 0.7898 | 0.97 | (0.76, 1.26) | 0.843 | |
| Non-valvular atrial fibrillation | | | | | | | | | | | | | 0.0048 |
| No | 20191 | 2955474 | 6.83 | 26390 | 2959822 | 8.92 | 1.3 | (1.28, 1.33)*** | <0.001 | 1.32 | (1.3, 1.35)*** | <0.001 | |
| Yes | 613 | 18310 | 33.48 | 718 | 19369 | 37.07 | 1.11 | (1, 1.24) | 0.0528 | 1.09 | (0.98, 1.22) | 0.1151 | |
| Chronic kidney disease | | | | | | | | | | | | | 0.0194 |
| No | 20002 | 2936951 | 6.81 | 26136 | 2940565 | 8.89 | 1.3 | (1.28, 1.33)*** | <0.001 | 1.32 | (1.3, 1.35)*** | <0.001 | |
| Yes | 802 | 36833 | 21.77 | 972 | 38626 | 25.16 | 1.16 | (1.06, 1.28)** | 0.0015 | 1.14 | (1.04, 1.25)** | 0.006 | |
| Chronic liver disease | | | | | | | | | | | | | 0.1956 |
| No | 20588 | 2957370 | 6.96 | 26838 | 2961228 | 9.06 | 1.3 | (1.28, 1.32)*** | <0.001 | 1.32 | (1.29, 1.34)*** | <0.001 | |
| Yes | 216 | 16414 | 13.16 | 270 | 17963 | 15.03 | 1.15 | (0.97, 1.38) | 0.1152 | 1.1 | (0.92, 1.32) | 0.3071 | |
| Peripheral vascular disease | | | | | | | | | | | | | 0.0004 |
| No | 20294 | 2946318 | 6.89 | 26556 | 2950598 | 9.00 | 1.31 | (1.28, 1.33)*** | <0.001 | 1.32 | (1.3, 1.35)*** | <0.001 | |
| Yes | 510 | 27466 | 18.57 | 552 | 28593 | 19.31 | 1.05 | (0.93, 1.18) | 0.447 | 1.03 | (0.92, 1.17) | 0.5951 | |
| Malignancy | | | | | | | | | | | | | <0.0001 |
| No | 19778 | 2880137 | 6.87 | 25525 | 2886159 | 8.84 | 1.29 | (1.26, 1.31)*** | <0.001 | 1.31 | (1.28, 1.33)*** | <0.001 | |
| Yes | 1026 | 93647 | 10.96 | 1583 | 93032 | 17.02 | 1.53 | (1.42, 1.66)*** | <0.001 | 1.51 | (1.4, 1.63)*** | <0.001 | |
| Venous thromboembolism | | | | | | | | | | | | | 0.2366 |
| No | 20668 | 2963941 | 6.97 | 26958 | 2969593 | 9.08 | 1.3 | (1.28, 1.32)*** | <0.001 | 1.32 | (1.29, 1.34)*** | <0.001 | |
| Yes | 136 | 9843 | 13.82 | 150 | 9598 | 15.63 | 1.13 | (0.9, 1.42) | 0.3045 | 1.01 | (0.8, 1.28) | 0.9209 | |
| Smoking | | | | | | | | | | | | | 0.1360 |
| No | 20656 | 2945236 | 7.01 | 26941 | 2957534 | 9.11 | 1.3 | (1.27, 1.32)*** | <0.001 | 1.32 | (1.29, 1.34)*** | <0.001 | |

(*Continued*)

**Table 3.** (Continued)

| Variables | Non-blepharitis | | | Blepharitis | | | Crude | | | Adjusted | | | |
|---|---|---|---|---|---|---|---|---|---|---|---|---|---|
| | n | PY | IR | n | PY | IR | cHR | CI | P | aHR† | CI | P | P for interaction |
| Yes | 148 | 28548 | 5.18 | 167 | 21657 | 7.71 | 1.5 | (1.2, 1.87)*** | <0.001 | 1.11 | (0.89, 1.38) | 0.3616 | |
| Obesity | | | | | | | | | | | | | 0.5408 |
| No | 20705 | 2957487 | 7.00 | 26972 | 2963202 | 9.10 | 1.3 | (1.28, 1.32)*** | <0.001 | 1.32 | (1.29, 1.34)*** | <0.001 | |
| Yes | 99 | 16297 | 6.07 | 136 | 15989 | 8.51 | 1.41 | (1.09, 1.82)** | 0.0097 | 1.36 | (1.05, 1.77)* | 0.0208 | |

PY, person-years; IR, incidence rate, per 10,000 person-years; cHR, crude hazard ratio; aHR, adjusted hazard ratio.

† Adjusted hazard ratio in multivariable analysis of sex, age, and comorbidities.

* $p < 0.05$

** $p < 0.01$

*** $p < 0.001$.

The event number of coronary artery disease was less than 3, the data cannot be retrieved according to the database guidelines.

## Discussion

To the best of our knowledge, this 10-year nationwide population-based cohort study is the first to present the epidemiologic association of blepharitis and IS. This study shows that patients with blepharitis have a 32% increased risk of IS, which is a potentially devastating neurological disease with a poor prognosis.

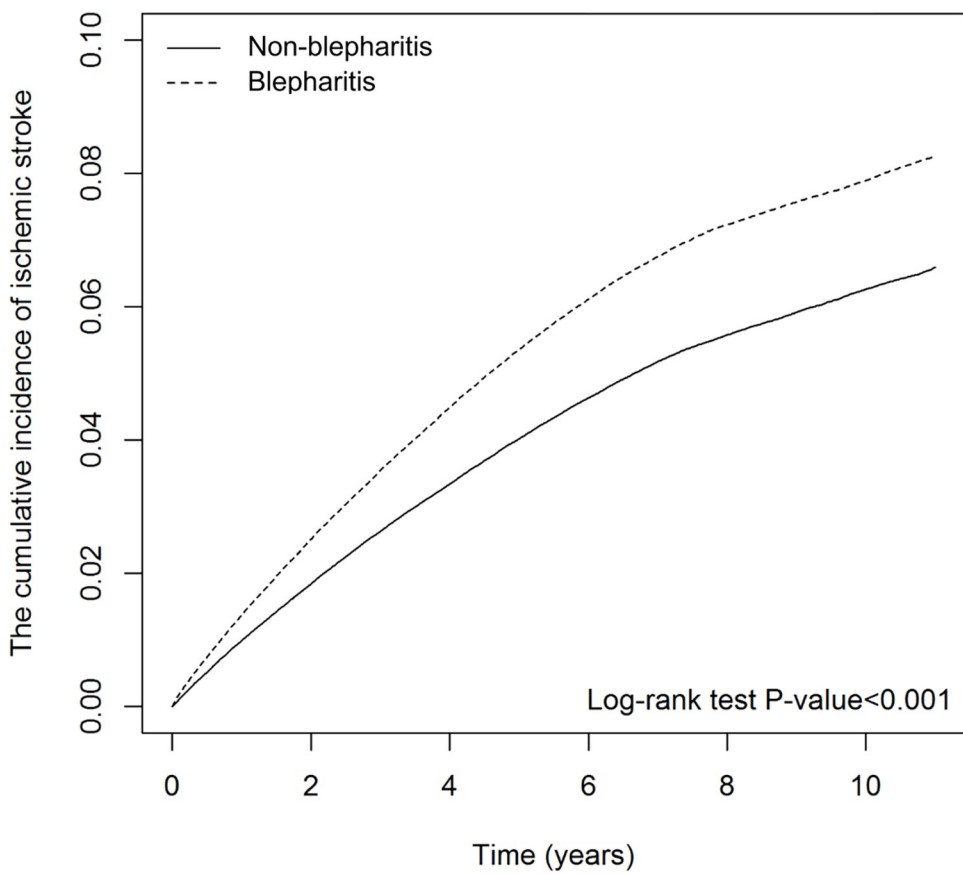

**Fig 2. The cumulative incidence curves of ischemic stroke estimated by cause-specific Cox proportional hazard regression model.**

**Table 4. The risks of ischemic stroke in the blepharitis cohort relative to the non-blepharitis cohort in terms of different follow-up time.**

|  | Non-blepharitis | | | Blepharitis | | | | | | |
|---|---|---|---|---|---|---|---|---|---|---|
| Years | n | PY | IR | n | PY | IR | cHR | (95% CI) | aHR[†] | (95% CI) |
| <1 | 4176 | 420063 | 9.94 | 5836 | 420354 | 13.88 | 1.4 | (1.34, 1.45)*** | 1.41 | (1.35, 1.46)*** |
| 1–3 | 6468 | 773348 | 8.36 | 8651 | 776483 | 11.14 | 1.33 | (1.29, 1.38)*** | 1.34 | (1.3, 1.38)*** |
| 4–6 | 6435 | 666459 | 9.66 | 8390 | 668735 | 12.55 | 1.28 | (1.23, 1.33)*** | 1.3 | (1.26, 1.34)*** |
| >6 | 3725 | 1113914 | 3.34 | 4231 | 1113620 | 3.80 | 1.14 | (1.09, 1.19)*** | 1.17 | (1.12, 1.22)*** |

PY, person-years; IR, incidence rate, per 10,000 person-years; cHR, crude hazard ratio; aHR, adjusted hazard ratio.

[†] Adjusted hazard ratio in multivariable analysis of sex, age, and comorbidities.

* $p < 0.05$

**$p < 0.01$

***$p < 0.001$

Blepharitis-associated risk factors can be divided into ocular and systemic factors; the former include topical medication and ocular surgery, and the latter include age, nutrition, testosterone, rosacea, Sjogren's syndrome, and systemic medication. The major risk factors of stroke include hypertension, DM, atrial fibrillation, and hyperlipidemia [17–19]. In a meta-analysis of women with DM, the risk of IS increased twofold, whereas the risk of intracranial hemorrhage increased slightly [20]. Almost all stroke burdens are attributable to modifiable risk factors worldwide [21]. Smoking, alcohol consumption, and air pollution are also likely casual risk factors [12, 22, 23]. Obesity, family history of stroke, atherosclerosis of the internal/common carotid artery, atrial fibrillation, coronary artery disease, transient ischemic attack, lower levels of high-density lipoproteins, aging, DM, and male sex [24] were significantly associated with IS [25].

To minimize potential confounding effect and assure convincing results with validity, we performed a comprehensive literature review and included comorbidities for adjustment of HR. Moreover, patients who had a history of disease that may be related to dry eye syndrome or diagnosis with keratoconjunctivitis during the study period were excluded. Because ocular surgery may pose harm to the corneal nerve and change the tear film, such as corneal incision during cataract surgery, patients who had undergone ocular surgery 4 weeks before the index date was excluded. Finally, patients using systemic medications (antihistamine, antiandrogen, 13-cis-retinoid acid, lipid-lowering agents, and β-blockers) and topical medications (ophthalmic glaucoma medications, steroids, epinephrine, anti-infectives, NSAIDs, antihistamines, anticholinergics, and artificial tears and lubricants) that may affect tearing or tear film composition were also excluded if any of them were prescribed 4 weeks before the index date.

Although the precise processes contributing to the link between blepharitis and IS remain unclear, we present putative pathways in terms of inflammation, infection, and secondary coagulopathy to explain the contribution of blepharitis to IS. First, in blepharitis, an inflammatory process that is limited to the orbit may indicate potential systemic inflammation. Macrophage and T-lymphocyte secretion of inflammatory mediators or growth factors, such as interleukins (ILs), transforming growth factor beta (TGF-β), and tumor necrosis factor alpha (TNF-α), enhances the inflammatory process. These cytokines at the inflammation site move via the blood circulation to the whole body, including the brain, and induce further immune response. Significant increase in the levels of IL-6 was observed in patients with IS following the ischemic event. In addition, a higher circulating IL-6 level is associated with an increased IS risk [26]. It plays a bidirectional role in the communication between leukocytes and vascular endothelium. IL-6 is recognized as an upstream inflammatory cytokine and accelerate the

downstream inflammation toward atherosclerosis [27]. The increased concentrations of IL-6 in the blood may also stimulate the production of acute-phase reactants, thus increasing blood coagulability. Matrix metallopeptidase 9 (MMP-9) concentration is also elevated in blepharitis and other ocular surface diseases [28, 29]. MMP-9 expression is also found to be elevated and may serve as a biomarker for acute IS [30].

Second, infection is a major contributor to the etiology of IS [31]. In patients with a history of prolonged systemic infection and inflammation, cerebral arteriosclerosis manifests with greater severity. In addition, dental infection was linked to cerebral aneurysm rupture [32]. Several proposed mechanisms have been elucidated, for example, human immunodeficiency virus, herpesviruses, and *Chlamydia pneumoniae* may accelerate atherosclerosis through the induction of TNF-α and IL-2 in response to antigens [33]. Herpes simplex virus, varicella zoster virus, and syphilis may directly invade the arterial wall and promote atheroma formation after being delivered to the arterial wall by circulating monocytes. Upon invasion of the endothelium, which initiates atherogenesis, a small percentage of bacterial pathogens can survive inside host cells for an extended period [33]. Furthermore, several mechanisms are associated with chronic infection and thrombosis. Proposed mechanisms include inflammation-induced thrombosis, impaired endothelial function, infection-induced platelet activation and aggregation, arrhythmias precipitated by infection leading to thrombosis, and dehydration status in sepsis-induced thrombosis [34]. Acute or chronic infection may even exaggerate the local or systemic inflammation response. Recurrent infections or intermittent reactivations of chronic infections may contribute to the exacerbations of atherosclerotic vasculopathy, even in minor systemic infections that could trigger acute IS in at-risk population with vasculopathy [35].

Polybacterial infections elevated the pathogen burden in another scale. Some organisms have been cultivated from atheromas, including *Staphylococcus epidermidis*, *Salmonella infantis*, and *Propionibacterium acnes*. Odontogenic viridans streptococci were detected in the thrombus of patients with acute IS [36]. Patients with blepharitis were found to have an increased burden of *Staphylococcus*, *Corynebacterium*, and *Enhydrobacter* infection [37]. During the course of chronic blepharitis, these pathogens may be circulated to the vascular wall of the central nervous system. Bacteremia is a powerful stimulus of inflammation and thrombosis. Sepsis was associated with 28-fold and 12-fold increased odds of IS [38]. The innate and adaptive immune systems, which respond to vascular injury, are also involved in the development of atherosclerosis. Endothelial activation, the release of growth factors, and monocyte adhesion/migration are triggered by the inflammatory process. Following the formation of foam cells, smooth muscle cells multiply. Atherogenesis was finally began and advanced. Nevertheless, the precise role of infection in cerebrovascular disease is highly complex and is still a matter of debate.

Third, inflammation and infection may induce coagulopathy. Seino et al. [39] reported the local expression of IL-6 in atherosclerotic plaques and atherosclerotic arteries. The level of IL-6 expression was 10 to 40 times higher than in normal tissues. IL-6 enhanced the formation and rupture of atherosclerotic plaques, hence accelerating the progression of atherosclerosis [40]. Additionally, IL-6 activates the coagulation cascade but does not affect fibrinolysis. In the acute setting, patients with sepsis secondary to infection had significantly elevated levels of IL-6 [41]. Adipose tissue may contribute to one-third of the total IL-6 in peripheral circulation and secrete TNF-α, and obesity is putative to resemble a low-grade pro-inflammatory state with burden of oxidative stress [42]. Higher level of ILs was also demonstrated on ocular surface. The risk of IS in patients with blepharitis cannot be explicated by common and rare major risk factors of IS, suggesting that blepharitis may be an independent risk factor of IS. Based on the findings, patients with blepharitis have an excess risk of IS (9.10 in 10,000 persons per year).

### Strengths and limitations

In this nationwide population-based study, specific strength is the 10-year observation period and the availability of considerable data on demographics and comorbidities that may influence results. We performed comprehensive work on consideration of any confounding effects before data analysis, including exclusion of the use of topical or systemic medications related to dry eye 4 weeks before index date to ensure adequate wash-out period.

However, this study has some limitations. First, blepharitis and IS were defined using diagnostic codes from the database; thus, misclassification because of coding errors or misdiagnosis might be an issue. As only diagnoses from at least two outpatient visits or one hospitalization record will be collected, it should be exceedingly uncommon. Second, blepharitis which did not gain medical attention for years, it is difficult to accurately identify truly incident cases based on medical records. Third, although substantial consideration of confounding factors, bias may still exist because of uncovered or unobserved ones. Fourth, the findings in this study should be cautiously interpreted given the retrospective design, which indicated the correlation between blepharitis and IS rather than casualty. Finally, the severity of blepharitis cannot be defined because data are not available on the NHIRD; thus, a subgroup analysis based on severity of blepharitis cannot be conducted. A prospective study investigating inflammatory markers and severity of blepharitis is warranted to validate the present findings though the knowledge of pro-inflammatory cytokines in blepharitis remain scarce.

### Conclusions

This preliminary study brings a new light on blepharitis. In patients with blepharitis, the risk of IS significantly increases and persists more than 6 years. Therefore, it was recommended that individuals with chronic or recurrent blepharitis undergo early management of blepharitis and control of inflammation or infection. It may raise awareness on blepharitis and inspire further investigation of the difference in the nature of IS by neurologist, and reiterate the need for consider the possibility of IS for patients with blepharitis by ophthalmologist. Further research on brain–eye axis is worthwhile.

### Author Contributions

**Conceptualization:** Jing-Xing Li.

**Data curation:** Jing-Xing Li, Ying-Hsiu Shih.

**Formal analysis:** Ying-Hsiu Shih.

**Methodology:** Jing-Xing Li.

**Supervision:** Chun-Chi Chiang.

**Validation:** Shu-Bai Hsu.

**Visualization:** Jing-Xing Li, Shu-Bai Hsu.

**Writing – original draft:** Jing-Xing Li.

**Writing – review & editing:** Jing-Xing Li, Shu-Bai Hsu, Yi-Yu Tsai, Ying-Hsuen Wu, You-Ling Li, Chun-Chi Chiang.

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
