## [Decision Letter · Decision Letter 0]

22 Nov 2022

PONE-D-22-29971The Risk of Ischemic Stroke Significantly Increases in Individuals with Blepharitis: A Population-based Study Involving 424,161 PatientsPLOS ONE

Dear Dr. Chiang,

Thank you for submitting your manuscript to PLOS ONE. After careful consideration, we feel that it has merit but does not fully meet PLOS ONE’s publication criteria as it currently stands. Therefore, we invite you to submit a revised version of the manuscript that addresses the points raised during the review process.

We look forward to receiving your revised manuscript.

Kind regards,

Redoy Ranjan, MBBS, MRCSEd, Ch.M., MS (CV&TS), FACS

Academic Editor

PLOS ONE

Journal Requirements:

Reviewers' comments:

Reviewer's Responses to Questions

**Comments to the Author**

1. Is the manuscript technically sound, and do the data support the conclusions?

Reviewer #1: Yes

Reviewer #2: Yes

2. Has the statistical analysis been performed appropriately and rigorously? 

Reviewer #1: Yes

Reviewer #2: Yes

3. Have the authors made all data underlying the findings in their manuscript fully available?

Reviewer #1: Yes

Reviewer #2: Yes

4. Is the manuscript presented in an intelligible fashion and written in standard English?

Reviewer #1: Yes

Reviewer #2: Yes

5. Review Comments to the Author

**Reviewer #1: **This pioneering study needs to be replicated but at a prospective design, with inclusion of surrogate markers like Il-6 and with severity of blepharitis included. the inclusion of odontogenic bacteria can be further done . A noteworthy citation that may aid in the future is this: Pyysalo MJ, Pyysalo LM, Pessi T, Karhunen PJ, Öhman JE. The connection between ruptured cerebral aneurysms and odontogenic bacteria. J Neurol Neurosurg Psychiatry. 2013 Nov;84(11):1214-8. doi: 10.1136/jnnp-2012-304635. Epub 2013 Jun 12. PMID: 23761916.

**Reviewer #2:** It is a new idea. however, what about the type of infarction? Is it large vessel or small vessel disease. In addition,is it thrombosis or embolic stroke. And what do you recommend after getting these results about the relation between blepharitis and stroke?

6. PLOS authors have the option to publish the peer review history of their article (what does this mean?). If published, this will include your full peer review and any attached files.

Reviewer #1: **Yes: **Mohamed Mostafa

Reviewer #2: No

---

## [Author Response · Author response to Decision Letter 0]

10 Dec 2022

Response to reviewer 1’s comment

Comment: This pioneering study needs to be replicated but at a prospective design, with inclusion of surrogate markers like Il-6 and with severity of blepharitis included. the inclusion of odontogenic bacteria can be further done . A noteworthy citation that may aid in the future is this: Pyysalo MJ, Pyysalo LM, Pessi T, Karhunen PJ, Öhman JE. The connection between ruptured cerebral aneurysms and odontogenic bacteria. J Neurol Neurosurg Psychiatry. 2013 Nov;84(11):1214-8. doi: 10.1136/jnnp-2012-304635. Epub 2013 Jun 12. PMID: 23761916.

Response: Thank you for your excellent suggestion on this topic. It has been observed that chronic infections are risk factors for ischemic stroke, and odontogenic bacteria are one of the established sources [1]. The research you mentioned supports the notion that dental infection is associated with cerebral vascular pathological changes. To corroborate the results of the present investigation, a prospective study comparing systemic/localized inflammatory markers such as IL-1 and IL-6 in blepharitis patients and those developing an ischemic stroke is required. Investigating the microbiology of tears and thrombus in patients with blepharitis may be helpful. Your valuable viewpoint is added to the Discussion section.

1. Patrakka, O., et al., Oral Bacterial Signatures in Cerebral Thrombi of Patients With Acute Ischemic Stroke Treated With Thrombectomy. J Am Heart Assoc, 2019. 8(11): p. e012330.

----------

Response to reviewer 2’s comment

Comment: It is a new idea. however, what about the type of infarction? Is it large vessel or small vessel disease. In addition, is it thrombosis or embolic stroke. And what do you recommend after getting these results about the relation between blepharitis and stroke?

Response: Thank you for your insightful comments. These are intriguing questions regarding the occluded arteries and type of ischemic stroke. But it was frustrating when we wanted to investigate the etiology further. Since this retrospective study defined study populations based on diagnostic codes, the information about location of culprit vessels and kind of ischemic stroke were not available in database. We concurred that these issues are crucial for validating the present findings. The origin of blood clots varies between thrombotic and embolic strokes. To minimize the potential bias, we adjusted the confounding factors such as non-valvular atrial fibrillation and venous thromboembolism in the analysis.

According to the findings of present study, we recommend early management of blepharitis. The risk of ischemic stroke persisted significantly higher in blepharitis cohorts than non-blepharitis cohorts for years according to the subgroup analysis. We hypothesized that chronic infection and inflammation played a crucial role in the relationship between blepharitis and ischemic stroke. Inflammatory cytokines may elevate immune response and pro-thrombotic states, hence stimulating thrombi development. We added additional detail in the Conclusions section.

---

## [Decision Letter · Decision Letter 1]

11 Jan 2023

PONE-D-22-29971R1The risk of ischemic stroke significantly increases in individuals with blepharitis: A population-based study involving 424,161 patientsPLOS ONE

Dear Dr. Chiang,

Thank you for submitting your manuscript to PLOS ONE. After careful consideration, we feel that it has merit but does not fully meet PLOS ONE’s publication criteria as it currently stands. Therefore, we invite you to submit a revised version of the manuscript that addresses the points raised during the review process.

ACADEMIC EDITOR: The authors are thanked for this submission to PLOS ONE. After a critical external peer review by three experts, I reinforce improving the clarity and presentation of your paper and adding recent references with in-text citations. Please see the attached reviewer comments detail below.

We look forward to receiving your revised manuscript.

Kind regards,

Redoy Ranjan, MBBS, MRCSEd, Ch.M., MS (CV&TS), FACS

Academic Editor

PLOS ONE

Reviewers' comments:

Reviewer's Responses to Questions

**Comments to the Author**

1. If the authors have adequately addressed your comments raised in a previous round of review and you feel that this manuscript is now acceptable for publication, you may indicate that here to bypass the “Comments to the Author” section, enter your conflict of interest statement in the “Confidential to Editor” section, and submit your "Accept" recommendation.

Reviewer #1: All comments have been addressed

Reviewer #2: All comments have been addressed

Reviewer #3: All comments have been addressed

2. Is the manuscript technically sound, and do the data support the conclusions?

Reviewer #1: Yes

Reviewer #2: Yes

Reviewer #3: Partly

3. Has the statistical analysis been performed appropriately and rigorously? 

Reviewer #1: Yes

Reviewer #2: Yes

Reviewer #3: Yes

4. Have the authors made all data underlying the findings in their manuscript fully available?

Reviewer #1: Yes

Reviewer #2: Yes

Reviewer #3: Yes

5. Is the manuscript presented in an intelligible fashion and written in standard English?

Reviewer #1: Yes

Reviewer #2: Yes

Reviewer #3: Yes

6. Review Comments to the Author

Reviewer #1: I would like to thank the honorable authors for including my comments in the manuscript and look forward to future prospective version of similar works.

Reviewer #2: Thanks for addressing the comments. And one last comment on the word" causal " as it is written "casual" in discussion section please correct.

Reviewer #3: 1．The authors statistically demonstrated an association between ischemic stroke (IS) and blepharitis. However, some points were difficult to understand, probably due to my lack of ability.

My understanding may be wrong, but, in this study, the subjects were all patients with NHIRD from 2008 to 2018 who were excluded from the exclusion criteria. Authors created two cohorts with or without blepharitis under the assessment of matching criteria, and examined the incidence of new IS in a follow-up up to 2019.

But, for example, the following 1) and 2) questions arise in the description of ‘Methods’, I think it is better to use Fig. 1 to organize and describe 'Methods' in an easy-to-understand manner.

1) Study Design:

① Are the following patients followed until Dec. 31, 2019? : Patients diagnosed with IS during the 10-year period between Jan. 1, 2008 and Dec. 31, 2018 among those diagnosed with blepharitis.

② Does the history of stroke, transient ischemia attack on line 6 mean past history before 2008?

2) Procedures:

The content of the blepharitis cohort and the non-blepharitis cohort on the third line is unclear. Is the blepharitis cohort the IS patients with blepharitis? Is the non-blepharitis cohort the IS patients without blepharitis? Alternatively, are the blepharitis cohorts the blepharitis patients other than the exclusion criteria patients? Are the non-blepharitis cohort patients with all diseases other than blepharitis and exclusion criteria patients?

2．More importantly, the definition of IS, which is the basis of this research, is unclear. Please clarify the following questions.

　The diagnosis of IS is made according to the ICD-9 and ICD-10 classifications, but the definition of IS in this study is unclear. ICD-9-CM 433 and 434 also include 'without mention of infarction'. ICD-10-CM I65 and I66 are 'not resulting in cerebral infarction'. It is necessary to clarify whether the IS criteria are infarction, include CBF reduction without infarction, or include arterial occlusion without CBF reduction. They will change the results and considerations.

3．The definition of 'IS' in the paper cited in ‘Discussion’ may be different from the definition of 'IS' in reference papers. For example 'IS' in reference [20] is ICD-10-CM I63, that is ‘cerebral infarction’.

4．In ‘Discussion’, the relationship between stroke risk factors and inflammatory processes is fully described using reference papers. However, there are few reference papers on the inflammatory process in blepharitis, and reference [32] is a small number patients (7 cases) study of chronic blepharitis. Reference [39] described the result of IL in uveitis, and not a study in blepharitis. It would be desirable to cite more reference papers on inflammatory processes in blepharitis and discuss the relationship between blepharitis and ischemic stroke via inflammatory processes more concisely.

5.　On line 13 of 'Strengths and Limitation', isn't 'can' in 'Finally, the severity of blepharitis can be defined' 'cannot'?

7. PLOS authors have the option to publish the peer review history of their article (what does this mean?). If published, this will include your full peer review and any attached files.

Reviewer #1: **Yes: **Mohamed Mostafa

Reviewer #2: No

Reviewer #3: No

---

## [Author Response · Author response to Decision Letter 1]

21 Mar 2023

PONE-D-22-29971R1

Reviewer #1: 

I would like to thank the honorable authors for including my comments in the manuscript and look forward to future prospective version of similar works.

A: Thank you for your great suggestions, we hope do it better in revision.

Reviewer #2: 

Thanks for addressing the comments. And one last comment on the word" causal " as it is written "casual" in discussion section please correct.

A: Thank you for your reminder and careful review, we correct this mistake.

Reviewer #3: 

1．The authors statistically demonstrated an association between ischemic stroke (IS) and blepharitis. However, some points were difficult to understand, probably due to my lack of ability.

My understanding may be wrong, but, in this study, the subjects were all patients with NHIRD from 2008 to 2018 who were excluded from the exclusion criteria. Authors created two cohorts with or without blepharitis under the assessment of matching criteria, and examined the incidence of new IS in a follow-up up to 2019. But, for example, the following 1) and 2) questions arise in the description of ‘Methods’, I think it is better to use Fig. 1 to organize and describe 'Methods' in an easy-to-understand manner.

A: We deeply apologize the inconvenience caused by the unclear description of the study's methodology. We have tried to make every effort to make Figure 1 more informative as below. After discussion with the statistician, he did not recommend that we alter the flowchart to some extent, as it involved the case selection and deletion procedure in R software. In order to avoid any confusion in the method section, we hope to elaborate on the study design and revise the original description. Thank you for your understanding. The old version of Figure 1 is on the left, and the updated one is on the right.

1) Study Design:

① Are the following patients followed until Dec. 31, 2019? Patients diagnosed with IS during the 10-year period between Jan. 1, 2008 and Dec. 31, 2018 among those diagnosed with blepharitis.

② Does the history of stroke, transient ischemia attack on line 6 mean past history before 2008?

A: (1) Yes, all patients included were followed until December 31, 2019, onset of IS, or withdrawal from NHIRD. Patients diagnosed with blepharitis instead of IS between January 1, 2008 and December 31, 2018 were included. (2) Yes, the patients with history of stroke, transient ischemia attack, graft-versus-host disease, human immunodeficiency virus infection, Sjögren’s syndrome, rheumatoid arthritis, or current keratoconjunctivitis before January 1, 2008 with be excluded. We add the date to exclusion criteria point 3.

2) Procedures:

The content of the blepharitis cohort and the non-blepharitis cohort on the third line is unclear. Is the blepharitis cohort the IS patients with blepharitis? Is the non-blepharitis cohort the IS patients without blepharitis? Alternatively, are the blepharitis cohorts the blepharitis patients other than the exclusion criteria patients? Are the non-blepharitis cohort patients with all diseases other than blepharitis and exclusion criteria patients?

A: We apologize for the confusion. These two cohorts were defined based solely on blepharitis status, independent of IS. (1) Blepharitis cohort included blepharitis patients. (2) The non-blepharitis cohort consists of patients who do not have blepharitis. (3)(4) The exclusion criteria definition for the blepharitis cohort and the non-blepharitis cohort were identical.

2．More importantly, the definition of IS, which is the basis of this research, is unclear. Please clarify the following questions. The diagnosis of IS is made according to the ICD-9 and ICD-10 classifications, but the definition of IS in this study is unclear. ICD-9-CM 433 and 434 also include 'without mention of infarction'. ICD-10-CM I65 and I66 are 'not resulting in cerebral infarction'. It is necessary to clarify whether the IS criteria are infarction, include CBF reduction without infarction, or include arterial occlusion without CBF reduction. They will change the results and considerations.

A: This is a crucial issue. The statistician who initially responded to this study has resigned and handed this study to a colleague. After careful confirmation by the new corresponding statistician, we assure that the following codes were utilized to retrieve data from the database (ICD-9-CM 433−437; ICD-10-CM I63, I67.89).

3．The definition of 'IS' in the paper cited in ‘Discussion’ may be different from the definition of 'IS' in reference papers. For example 'IS' in reference [20] is ICD-10-CM I63, that is ‘cerebral infarction’.

A: The sensitivity of ischemic stroke is 82% with these diagnostic codes (ICD-9-CM 434; ICD-10-CM I63 or ICD-9-CM 434, 436).1 A study by Sun et al.2 use diagnostic codes ICD-9-CM 433 (occlusion and stenosis of precerebral arteries), 434 (occlusion of cerebral arteries), 436 (acute, but ill-defined, cerebrovascular disease) for ischemic stroke. In previous published works3-5 from our research team, ICD-9-CM 433−438 were adopted for ischemic stroke. In this study, we used ICD-9-CM 433−437 and omitted ICD-9-CM 438, because ICD-9-CM 438 refers to complications associated with cerebrovascular disease, which did not specify ischemic stroke. All references were listed at the bottom of the rebuttal letter.

4．In ‘Discussion’, the relationship between stroke risk factors and inflammatory processes is fully described using reference papers. However, there are few reference papers on the inflammatory process in blepharitis, and reference [32] is a small number patients (7 cases) study of chronic blepharitis. Reference [39] described the result of IL in uveitis, and not a study in blepharitis. It would be desirable to cite more reference papers on inflammatory processes in blepharitis and discuss the relationship between blepharitis and ischemic stroke via inflammatory processes more concisely.

A: This is an excellent suggestion; we remove [39] and the sentence. It is our duty to expand the information on blepharitis and ischemic stroke. After thorough review of relevant literatures, we confirmed that the pathogenic mechanisms driving the inflammatory response remain poorly understood over two centuries. In stroke, pro-inflammatory cytokines have been extensively studied, but not in blepharitis. TNF-α, IL-1β, IL-6, and IL-10 are inflammatory cytokines associated with IS that have been implicated as therapeutic targets and prognostic biomarkers. Intriguingly, matrix metalloproteinase-9 levels are elevated in both blepharitis and IS. Additional research is required in the field. We regret that we cannot provide a complete and definite response to this question.

5.　On line 13 of 'Strengths and Limitation', isn't 'can' in 'Finally, the severity of blepharitis can be defined' 'cannot'?

A: We correct the typo. Thank you.

Reference

1. McCormick N, Bhole V, Lacaille D, et al. Validity of Diagnostic Codes for Acute Stroke in Administrative Databases: A Systematic Review. PLoS One 2015;10(8):e0135834. doi: 10.1371/journal.pone.0135834 [published Online First: 2015/08/21]

2. Sun Y, Lee SH, Heng BH, et al. 5-year survival and rehospitalization due to stroke recurrence among patients with hemorrhagic or ischemic strokes in Singapore. BMC Neurol 2013;13:133. doi: 10.1186/1471-2377-13-133 [published Online First: 2013/10/04]

3. Lin CW, Chen WK, Hung DZ, et al. Association between ischemic stroke and carbon monoxide poisoning: A population-based retrospective cohort analysis. Eur J Intern Med 2016;29:65-70. doi: 10.1016/j.ejim.2015.11.025 [published Online First: 2015/12/26]

4. Huang WS, Tseng CH, Chen PC, et al. Inflammatory bowel diseases increase future ischemic stroke risk: a Taiwanese population-based retrospective cohort study. Eur J Intern Med 2014;25(6):561-5. doi: 10.1016/j.ejim.2014.05.009 [published Online First: 2014/06/08]

5. Fang CW, Tseng CH, Wu SC, et al. Association of Vagotomy and Decreased Risk of Subsequent Ischemic Stroke in Complicated Peptic Ulcer Patients: an Asian Population Study. World J Surg 2017;41(12):3171-79. doi: 10.1007/s00268-017-4127-z [published Online First: 2017/07/21]

---

## [Decision Letter · Decision Letter 2]

5 Apr 2023

The risk of ischemic stroke significantly increases in individuals with blepharitis: A population-based study involving 424,161 patients

PONE-D-22-29971R2

Dear Dr. Chiang,

We’re pleased to inform you that your manuscript has been judged scientifically suitable for publication and will be formally accepted for publication once it meets all outstanding technical requirements.

Kind regards,

Redoy Ranjan, MBBS, MRCSEd, Ch.M., MS (CV&TS), FACS

Academic Editor

PLOS ONE

Additional Editor Comments (optional):

**Review Comments to the Author**

Reviewer #1: Thank you for the review opportunity. I would like to see reproducible articles including intracerebral aneurysm/ hemorrhagic infarction.

Reviewer #3: The authors kindly answered my questions to compensate for my lack of understanding. And they accepted some of my comments and revised part of their manuscript. I greatly appreciate the efforts of the authors.

---

## [Editor Report · Acceptance letter]

19 Apr 2023

PONE-D-22-29971R2 

The risk of ischemic stroke significantly increases in individuals with blepharitis: A population-based study involving 424,161 patients 

Dear Dr. Chiang:

I'm pleased to inform you that your manuscript has been deemed suitable for publication in PLOS ONE. Congratulations! Your manuscript is now with our production department. 

Kind regards, 

on behalf of

Dr. Redoy Ranjan 

Academic Editor

PLOS ONE